# Enhancing nonlinear transcriptome- and proteome-wide association studies via trait imputation with applications to Alzheimer's disease

**Ruoyu He**[1,2☯], **Jingchen Ren**[1,2☯], **Mykhaylo M. Malakhov**[2], **Wei Pan**[2]*

**1** School of Statistics, University of Minnesota, Minneapolis, Minnesota, United States of America,
**2** Division of Biostatistics and Health Data Science, School of Public Health, University of Minnesota, Minneapolis, Minnesota, United States of America

☯ These authors also contributed equally to this work.

* panxx014@umn.edu

**Data availability statement:** EADB GWAS summary statistics data are publicly available from the European Bioinformatics Institute (EBI)

## Abstract

Genome-wide association studies (GWAS) performed on large cohort and biobank datasets have identified many genetic loci associated with Alzheimer's disease (AD). However, the younger demographic of biobank participants relative to the typical age of late-onset AD has resulted in an insufficient number of AD cases, limiting the statistical power of GWAS and any downstream analyses. To mitigate this limitation, several trait imputation methods have been proposed to impute the expected future AD status of individuals who may not have yet developed the disease. This paper explores the use of imputed AD status in nonlinear transcriptome/proteome-wide association studies (TWAS/PWAS) to identify genes and proteins whose genetically regulated expression is associated with AD risk. In particular, we considered the TWAS/PWAS method DeLIVR, which utilizes deep learning to model the nonlinear effects of expression on disease. We trained transcriptome and proteome imputation models for DeLIVR on data from the Genotype-Tissue Expression (GTEx) Project and the UK Biobank (UKB), respectively, with imputed AD status in UKB participants as the outcome. Next, we performed hypothesis testing for the DeLIVR models using clinically diagnosed AD cases from the Alzheimer's Disease Sequencing Project (ADSP). Our results demonstrate that nonlinear TWAS/PWAS trained with imputed AD outcomes successfully identifies known and putative AD risk genes and proteins. Notably, we found that training with imputed outcomes can increase statistical power without inflating false positives, enabling the discovery of molecular exposures with potentially nonlinear effects on neurodegeneration.

GWAS Catalog (https://www.ebi.ac.uk/gwas/) under accession no. GCST90027158. IGAP GWAS summary statistics data are publicly available from the EBI GWAS Catalog under accession no. GCST007511. Individual-level data from the UK Biobank (https://www.ukbiobank.ac.uk/), the Genotype-Tissue Expression (https://gtexportal.org/home/) Project, and the Alzheimer's Disease Sequencing Project (https://adsp.niagads.org/) are available by application through their respective data access processes. The code used for the analyses presented in this paper is available at https://github.com/RuoyuHe/LS-imputation_TWAS. The code for DeLIVR is available at https://github.com/RuoyuHe/DeLIVR. The code for LS-imputation is available at https://github.com/ren328/LSimputing. The code for PRS-CS is available at https://github.com/getian107/PRScs. LDpred2 is supplied with the R package "bigsnpr" (https://github.com/privefl/bigsnpr).

**Funding:** This work was supported by the National Institutes of Health (NIH) under grants U01 AG073079 (RH, JR, WP), RF1 AG067924 (MM, WP) and R01 HL116720 (MM, WP). The funders had no role in study design, data collection and analysis, decision to publish, or preparation of the manuscript.

**Competing interests:** The authors have declared that no competing interests exist.

## Author summary

Transcriptome-wide association studies (TWAS) and proteome-wide association studies (PWAS) are useful for identifying causal genes and proteins for complex human traits. However, the power of TWAS/PWAS to identify genes and proteins for late-onset diseases, such as Alzheimer's disease (AD), is limited by the small numbers of disease cases in some biobanks, largely due to the relatively young age of study participants. Traditional TWAS methods can overcome this limitation by relying on external genome-wide association study (GWAS) summary statistics, but they fail to capture nonlinear associations, which are particularly important in complex diseases such as AD. The main contribution of this paper is to demonstrate that by incorporating a newly proposed trait imputation method, LS-imputation, along with the widely used proxy AD imputation method, we can detect potentially nonlinear associations between molecular exposures and AD status. This approach enhances the power of nonlinear TWAS/PWAS for studying AD and possibly other late-onset diseases. Furthermore, we show that applying LS-imputation within the TWAS/PWAS framework is unlikely to lead to false discoveries, supporting its reliability in genetic studies of AD.

## 1. Introduction

Alzheimer's disease (AD), a complex polygenic neurodegenerative disorder and the most prevalent form of dementia, has captured significant attention within the genetics research community. Genetic mutations are recognized as a predominant factor in AD pathogenesis, with heritability estimates for late-onset AD ranging from 60% to 80% [1]. The advent of large biobanks and the development of genome-wide association studies (GWAS) have accelerated the identification of single-nucleotide polymorphisms (SNPs) and genetic loci linked to AD. In the past decade alone, the number of known GWAS loci significantly associated with AD has increased from 20 to over 90 [2–7]. Although these findings have significantly advanced our understanding of the genetic architecture of AD, statistical power for studying neurodegenerative conditions may plateau despite ever-larger sample sizes due to the demographic makeup of most large biobanks. Large studies such as the UK Biobank (UKB) primarily enroll relatively young individuals who are healthier than the general population [8,9], so the number of AD cases among biobank participants does not grow at the same rate as the overall sample size. Thus, methods for imputing the disease phenotypes of participants who are likely to develop AD as they age are necessary in order to refine research outcomes and increase statistical power.

A common practice for addressing this issue is to use the GWAS-by-proxy (GWAX) method [10], which aggregates the family history of biobank participants to impute their expected AD status, known as "proxy AD." Since its proposal, GWAX has gained rapid popularity, with almost every recent AD GWAS analysis adopting this method to enhance statistical power [3,4,11–14]. Although not commonly applied in this context, polygenic score methods can also be used to impute AD risk. Recently, another phenotype imputation method, LS-imputation, has been proposed [15,16]. Instead of borrowing information from family history, LS-imputation relies on published GWAS summary statistics to impute phenotypes for individuals with known genome-wide genotypes. Importantly, despite relying solely on GWAS summary statistics, LS-imputation has been shown to effectively reconstruct both linear and nonlinear genotype-phenotype associations, unlocking new possibilities for downstream analyses.

Testing for nonlinear genotype-phenotype associations has drawn growing interest in recent years [17,18], but such discoveries can be constrained by the limited sample size available for certain traits such as AD. By leveraging the capabilities of LS-imputation and proxy phenotypes to retain nonlinear genotype-phenotype relationships, the large sample sizes of modern biobanks now enable nonlinear transcriptome-wide association study (TWAS) inference [19–25]. More generally, this approach can be used for nonlinear association studies of any molecular traits [26–28], including proteome-wide association studies (PWAS) of plasma protein concentrations [29–32].

In this study, we evaluated the ability of various AD imputation methods to recapture genotype-phenotype relationships and compared their performance to that of using only observed trait in TWAS/PWAS inference tasks. In stage 1 of TWAS, we built prediction models using gene expression data from the Genotype-Tissue Expression (GTEx) Project version 8 [33]. Analogously, we also trained prediction models for protein expression using data from the UKB Pharma Proteomics Project [34,35]. Then we imputed gene/protein expression into the UKB data. In stage 2, we used the imputed expression as input and trained the DeLIVR model [36] using the imputed AD status of UKB individuals. Lastly, we performed association testing with diagnosed AD case/control phenotypes from the Alzheimer's Disease Sequencing Project (ADSP) release 4. Note that hypothesis testing was done on an independent dataset (ADSP) with clinically diagnosed AD cases in order to avoid potential pitfalls of testing with imputed data. Our nonlinear TWAS analysis identified known AD risk genes and proteins, as well as new putative targets. Notably, we found that training DeLIVR models using imputed AD outcomes resulted in the identification of more genes and proteins than when training solely using observed outcomes. Moreover, the distinct sets of results obtained from different imputation methods suggest that considering these methods together may provide a more complete picture of the genetic underpinnings of AD.

Additional analyses were also performed using UKB data for high-density lipoprotein (HDL) cholesterol to compare the Type I error rate and power of DeLIVR trained with an LS-imputed trait. Our results showed that DeLIVR trained with LS-imputed HDL cholesterol levels and tested on an independent dataset with observed HDL cholesterol levels could control the Type I error rate at a nominal level. In addition, training DeLIVR with the imputed trait often had higher power compared to training it on a smaller dataset with the observed trait.

## 2. Materials and methods

### 2.1. Overview of TWAS/PWAS

We begin by outlining the TWAS/PWAS framework. For conciseness, we will use TWAS as an example, though the same methods are also applicable to proteomics and other types of molecular traits. The causal model for TWAS is illustrated in Fig 1a. Formally, denote $Y \in \mathbb{R}^{n \times 1}$ to be the outcome trait with $n$ samples, $X \in \mathbb{R}^{n \times 1}$ to be a gene's or protein's expression levels, and $Z \in \mathbb{R}^{n \times m}$ to be the genotype matrix with $m$ genetic variants. The TWAS model is as follows,

$$X = Z\beta + \epsilon_1, \tag{1}$$

$$Y = g(X) + \epsilon_2, \tag{2}$$

where $g(X)$ is the target function of expression that we are interested in testing, and $\epsilon_1$ and $\epsilon_2$ are independent error terms. Traditionally, TWAS uses a reference dataset to estimate the stage 1 model (Eq 1) and another independent dataset to estimate the stage 2 model (Eq 2).

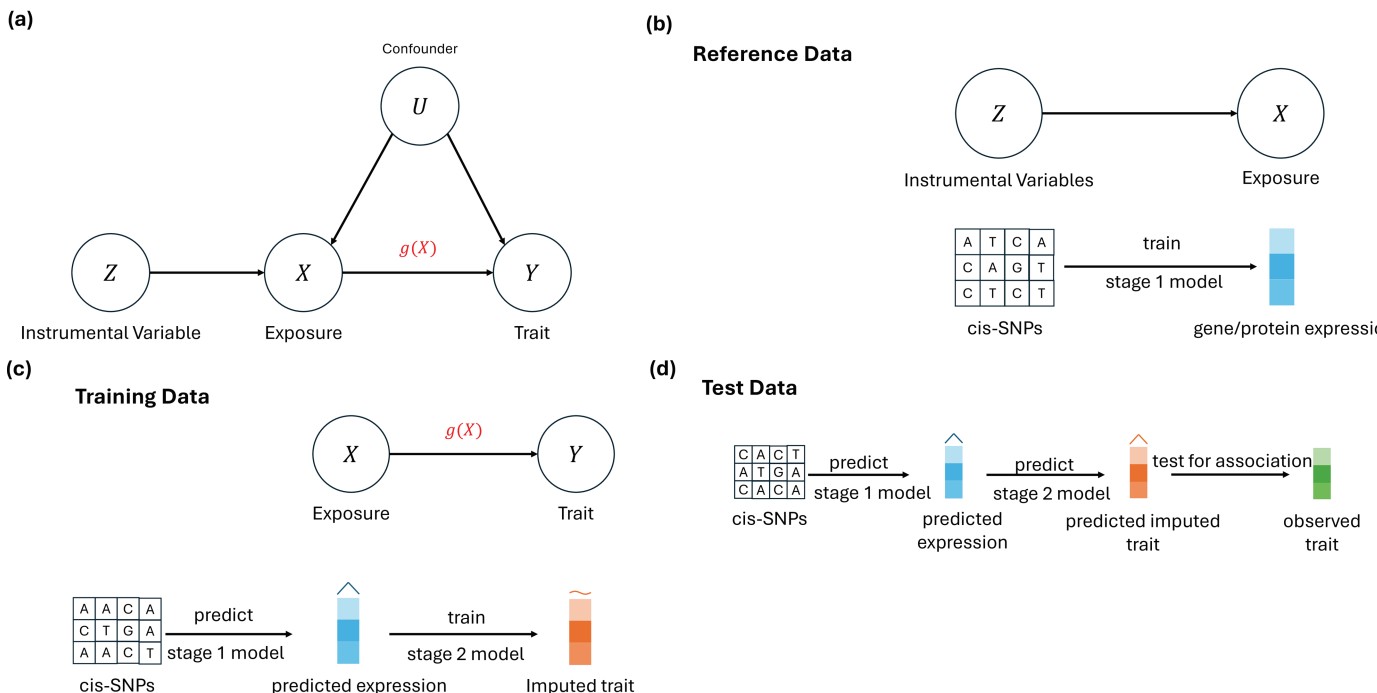

**Fig 1. Illustration of the TWAS/PWAS workflow. (a)** Traditional TWAS trains a model to predict an exposure ($X$) from genetic instruments ($Z$) and then tests the relationship between the predicted exposure and an outcome trait ($Y$). **(b)** The first stage of our TWAS framework is the same as in traditional TWAS, where a model is trained to predict gene or protein expression from local genetic variation. **(c)** Unlike traditional TWAS, which directly tests for association between predicted exposures and the outcome trait, our framework first trains a stage 2 model to predict the outcome trait. The outcome trait used for training may itself be imputed. **(d)** Hypothesis testing is performed on an independent test dataset. We use the stage 1 model to predict expression from genotypes, then we use the stage 2 model to predict the outcome trait from the predicted expression, and finally we test for association between the predicted outcome trait and its observed values.

Our framework differs from the traditional approach in that it requires three distinct datasets. First, we need a reference dataset containing genotype data and observed gene/protein expression data to build the stage 1 prediction model (Eq 1) for gene/protein expression using genetic variants as predictors (Fig 1b). Second, we require a large biobank dataset with genotype data but without phenotypes. We apply a trait imputation method to impute the outcome trait into this biobank dataset, and use the trained stage 1 model to predict gene/protein expression for those same individuals. Then we train the stage 2 model (Eq 2) using the predicted gene/protein expression as input and the imputed trait as the output (Fig 1c). Lastly, we need an independent test dataset with observed genotype and outcome trait data. We predict gene/protein expression on the test dataset using the stage 1 model and then predict the genetic component of the outcome trait using the stage 2 model. Hypothesis testing is performed to test the association between the predicted outcome trait and its observed values (Fig 1d). Unlike the traditional TWAS approach, our framework performs hypothesis testing on an independent dataset with observed traits, rather than on the imputed traits themselves. This is crucial as testing with observed traits avoids any potential biases and uncertainties associated with imputed traits. In particular, any biases in the trait imputation process can affect downstream association tests if the imputed values are used as if they were observed (e.g. for hypothesis testing). An independent test dataset with observed traits provides an unbiased dataset against which the predictions can be validated. The association is then evaluated in a context that is free from any imputation biases. Furthermore, imputed

traits inherently carry estimation uncertainty, but testing on an independent dataset with observed traits ensures that the variability in the data is accurately captured, leading to valid inference. Theoretical justification for this framework is provided in [36].

**2.1.1. DeLIVR**  A variety of methods are available to estimate the TWAS model introduced in Eqs 1 and 2. In this study we primarily focus on the recently published DeLIVR method, which uses neural networks to estimate $E(Y|Z)$ nonparametrically [36]. Stage 1 of DeLIVR is the same as in standard TWAS, where we regress the gene expression $X$ on the genotypes $Z$ to obtain $\widehat{X} = Z\widehat{\beta}$. In stage 2, DeLIVR estimates and performs inference on $E(Y|Z) = E(g(X)|Z)$ without explicitly learning $g(X)$. In [36], the authors showed that under some assumptions, estimating $E(g(X)|Z)$ is sufficient for testing the association between the trait and the predicted expression levels. Furthermore, this approach resulted in a much more stable estimate compared to other deep learning methods for instrumental variables regression, such as DeepIV, and consequently yielded higher statistical power in hypothesis testing [36,37].

Let $\widehat{h}_\theta$ be a neural network parameterized by $\theta$ for estimating $E(g(X)|Z)$ with $\widehat{X} = Z\widehat{\beta}$. We solve for $\theta$ by minimizing the following loss,

$$L(Y, Z, \theta) = \frac{1}{n} \sum_{i=1}^{n} (Y_i - \widehat{h}_\theta(\widehat{X}_i))^2 + \lambda \|\theta\|_2^2, \tag{3}$$

where the last term is the ridge penalty.

**2.1.2. TWAS-L and TWAS-LQ**  We also consider the standard linear TWAS (TWAS-L) and a parametric nonlinear model proposed in [38] (TWAS-LQ) to compare with DeLIVR. Assuming $Y$, $X$, and $Z$ are standardized to have mean 0 and variance 1, we fit the following two stage 1 models: $X = Z\beta + \epsilon_1$ and $X^2 = Z\beta_2 + \epsilon_2$. Using the fitted models, we obtain $\widehat{X} = Z\widehat{\beta}$ and $\widehat{X^2} = Z\widehat{\beta}_2$. In stage 2 of TWAS-L, we fit $Y = \widehat{X}\theta + \epsilon$. In stage 2 of TWAS-LQ, we instead fit $Y = \widehat{X}\theta_1 + \widehat{X^2}\theta_2 + \epsilon$.

**2.1.3. Hypothesis testing**  Assuming that the unknown parameters in $\widehat{h}_\theta$ has been estimated by DeLIVR with a training dataset, we use an independent test set to calculate $\widehat{E}(g(X)|Z) = \widehat{h}_\theta(\widehat{X}$ and perform the following two tests in TWAS.

- **Global test:** We fit

$$Y = \alpha + \beta \, \widehat{E}(g(X)|Z) + \epsilon_G, \tag{4}$$

where $\epsilon_G \sim N(\mathbf{0}, \sigma_G^2 I_n)$ is an independent error term. Then we test the hypothesis $H_0 : \beta = 0$ vs. $H_1 : \beta \neq 0$.

- **Nonlinearity test**: We fit

$$Y = \alpha + \beta_1 \, \widehat{E}(X|Z) + \beta_2 \, \widehat{E}(g(X)|Z) + \epsilon_{NL}, \tag{5}$$

where $\epsilon_{NL} \sim N(\mathbf{0}, \sigma_{NL}^2 I_n)$ is an independent error term. Then we test the hypothesis $H_0 : \beta_2 = 0$ vs. $H_1 : \beta_2 \neq 0$.

For a given gene, we could aggregate the test results from different methods by simply combining their $p$-values using the Cauchy combination test [39], which may further improve power over that of a single model.

## 2.2. Trait imputation methods

In this section we summarize the methods we considered for imputing AD status. The imputed outcome trait values obtained from these methods were then subsequently used to train the stage 2 models for TWAS/PWAS, as described above.

**2.2.1. LS-imputation** LS-imputation leverages GWAS summary statistics from an independent study to impute disease phenotypes [15,16]. Suppose we have a GWAS summary (training) dataset $\{(\widehat{\beta}_j^*, \widehat{\sigma}_j^*) : j = 1, ..., p\}$ with $p$ genetic variants. For a new individual-level (test) dataset in which we only have genotypes $X$ with sample size $n_2$, we can impute the missing trait values $Y$ for these $n_2$ individuals by solving the following optimization problem,

$$\widetilde{Y} = \arg\min_Y \|\widehat{\beta}^* - \frac{1}{n_2 - 1} X^T Y\|^2 = (n_2 - 1)(XX^T)^+ X\widehat{\beta}^* = (n_2 - 1)(X^T)^+ \widehat{\beta}^*, \qquad (6)$$

where $(\cdot)^+$ denotes the Moore-Penrose generalized inverse. The objective is to find $\widetilde{Y}$ such that its correlation with the genotypes closely matches observed correlations in an independent GWAS from the same population. Due to computational constraints when dealing with large matrices, particularly in cases where both $p$ and $n_2$ are large, LS-imputation is applied in smaller batches of size $m$ [16]. The batch size $m$ is chosen to balance a bias-variance trade-off; smaller values of $m$ typically result in higher bias but lower variance in the imputed $Y$. In practice, we try a few values of $m$ and choose the one that yields marginal association results similar to those obtained from the external GWAS.

It is important to note that the LS-imputation method is based on ordinary least squares (OLS) estimation and is only directly applicable to traits where the GWAS summary statistics were obtained from a linear regression model for each variant. However, for a binary trait like AD, the GWAS summary statistics are typically obtained from logistic regression. To employ the LS-imputation method in this scenario, we first use the following approximation formulas to convert GLM-based summary statistics to OLS-based summary statistics [40],

$$\widehat{\beta} = \frac{e^{-\widehat{b}_0}}{(1 + e^{-\widehat{b}_0})^2} \widehat{b}_1, \qquad (7)$$

$$\mathrm{SE}(\widehat{\beta}) \approx \frac{e^{-\widehat{b}_0}}{(1 + e^{-\widehat{b}_0})^2} \mathrm{SE}(\widehat{b}_1), \qquad (8)$$

where $\widehat{b}_0$ is the estimated log odds of an individual with the non-counted allele being a diseased case and $\widehat{b}_1$ is the GLM-based effect size.

**2.2.2. Proxy AD** To counteract the underrepresentation of late-onset diseases such as AD in large biobanks such as the UKB, where cohorts predominantly consist of middle-aged individuals, a novel methodology known as GWAS-by-proxy (GWAX) was introduced by [10]. They constructed proxy AD cases by leveraging the health history information of the biobank participants' parents. Since its introduction, GWAX has gained increasing popularity. The majority of recently published AD GWAS studies are meta-analyses that merge associations computed from clinically diagnosed AD cases with associations computed from GWAX proxy cases in order to enhance both sample size and statistical power [3,4,11–13]. The adoption of GWAX has sparked many discussions and led to the development of various related methods for constructing AD proxies [4,41].

In our analyses, we explored two distinct strategies for constructing proxy AD cases from individuals in the UKB. The first strategy, as employed in [13], designates a participant as an AD proxy case if they reported that at least one biological relative (parent or sibling) was

diagnosed with AD. Participants who responded "Do not know" or "Prefer not to answer" when asked about their parents' AD history are excluded. In this strategy no distinction is made between proxy AD cases and clinically diagnosed AD cases during analysis, i.e., both are together considered to be AD cases. We refer to this imputation strategy as "AD Proxy."

The second strategy, as outlined in [4], involves evaluating whether participants' biological parents have a history of AD and taking into account the parents' current ages or ages at death. The proxy phenotype is then constructed as a linear score between 0 and 2 based on these factors. Each parent with a diagnosis of AD contributes 1 unit to the score, while each unaffected parent contributes the fraction (100 – age)/100, where "age" is the current age of the parent if they are still alive or the parent's age at death. The contribution for unaffected parents is capped at 0.32, which corresponds to the maximum population prevalence of AD. Participants with clinically diagnosed AD are given the maximum risk score of 2. Similarly to the first strategy, participants who responded "Do not know" or "Prefer not to answer" regarding their parents' disease status or age are excluded. We refer to this imputation strategy as "AD Proxy2."

**2.2.3. PRS-CS**  PRS-CS builds polygenic risk scores (PRS) by employing a high-dimensional Bayesian regression framework that incorporates a continuous shrinkage prior on genetic variant effect sizes [42]. In this paper, we used the Python implementation of PRS-CS available at https://github.com/getian107/PRScs. In particular, we applied PRS-CS-auto and used the provided European UKB reference panel. The risk scores computed by PRS-CS were directly used as the imputed trait values.

**2.2.4. LDpred2**  LDpred2 is another widely used method for constructing PRS, which leverages GWAS summary statistics and a linkage disequilibrium (LD) matrix [43]. LDpred2 is an enhanced and more efficient version of the original LDpred method [44]. In our analyses, we utilized LDpred2-auto as implemented in the R package bigsnpr (https://github.com/privefl/bigsnpr). Similarly to our application of PRS-CS, the risk scores computed by LDpred2 were directly used as the imputed trait values.

### 2.3. Imputation data sources

In this section we detail the datasets we used to facilitate trait imputation and the preprocessing steps we applied to each one. Recall that LS-imputation, PRS-CS, and LDpred2 require summary-level data from an external GWAS in addition to individual-level genotype data on the individuals for whom trait imputation is to be performed. We used summary-level AD GWAS data from either the EADB consortium or the IGAP consortium, and individual-level genotype data from the UKB. Imputation using AD Proxy and AD Proxy2, on the other hand, is only based on family history. For those methods we used individual-level data from the UKB. Note that the GWAS studies described in this section were only used for trait imputation, and they should not be confused with the biobank data used in stage 2 of the TWAS/PWAS workflow.

**2.3.1. EADB GWAS summary data**  The European Alzheimer & Dementia Biobank (EADB) consortium aggregates data from various European GWAS consortia focused on AD [13]. The EADB GWAS results were meta-analyzed with a GWAS-by-proxy conducted on the UKB dataset, following the AD Proxy strategy described in Sect 2.2.2, with a total of 21,101,114 genetic variants. The meta-analyzed EADB stage 1 GWAS included 85,934 AD cases (39,106 clinically diagnosed AD cases and 46,828 proxy AD cases) and 401,577 controls, for a total of 485,711 individuals.

**2.3.2. IGAP GWAS summary data**  The International Genomics of Alzheimer's Project (IGAP) is a large three-stage study based on GWAS analyses of individuals with European

ancestry [5]. The IGAP stage 1 meta-analysis included 11,480,632 SNPs in 21,982 AD cases and 41,944 cognitively normal controls from four consortia: the Alzheimer Disease Genetics Consortium (ADGC); the European Alzheimer's Disease Initiative (EADI); the Cohorts for Heart and Aging Research in Genomic Epidemiology Consortium (CHARGE); and the Genetic and Environmental Risk in Alzheimer's Disease/Defining Genetic, Polygenic and Environmental Risk for Alzheimer's Disease Consortium (GERAD/PERADES). In the IGAP stage 2 meta-analysis, 11,632 SNPs were genotyped and tested for association in an independent set of 8,362 AD cases and 10,483 controls. Meta-analysis of variants selected for analysis in stage 3A ($n = 11,666$) or stage 3B ($n = 30,511$) brought the final sample size to 35,274 clinical or autopsy-documented AD cases and 59,163 controls.

**2.3.3. UKB individual-level data** We imputed the AD status of 367,182 self-identified White British UKB individuals [34] using each of the trait imputation methods described in Sect 2.2. For the GWAS-based trait imputation methods, we utilized imputed UKB genotypic data along with EADB or IGAP GWAS summary data. For each of the two AD GWAS datasets, we first extracted all of the genetic variants in common between the GWAS and UKB data. Subsequently, we filtered out variants with a minor allele frequency (MAF) less than 0.05, missing rates larger than 10%, and those that failed a Hardy-Weinberg equilibrium exact test with $p$-values less than 0.001. Furthermore, we pruned out variants in high LD with a window size of 50 base pairs (bp), a step size of 1 bp, and an $r^2$ threshold of 0.8. For the EADB GWAS dataset, we ended up with 1,035,821 high-quality SNPs in common with UKB data. A total of 76,417 SNPs had $p$-values less than 0.05, from which we randomly selected 70,000 SNPs to use in the genotype-based trait imputation methods LS-imputation, PRS-CS, and LDpred2. For the IGAP GWAS dataset, we ended up with 1,053,329 high-quality SNPs in common with UKB data. A total of 63,094 SNPs had $p$-values less than 0.05, from which we randomly selected 60,000 SNPs to use in the genotype-based trait imputation methods.

Recall that LS-imputation requires splitting the individuals into smaller batches and performing the imputation batch-by-batch. For EADB data, we considered batch sizes of 40,000, 50,000, and 60,000. For IGAP data, we considered batch sizes of 30,000, 40,000, and 50,000.

For the family-based trait imputation methods, we obtained diagnosed AD cases from self-reported ICD10 diagnoses (data field 41270) and ICD10 cause of death (data fields 40001 and 40002). Our AD Proxy analyses included a total of 367,182 UKB individuals, comprising 313,294 controls and 53,888 cases (51,815 AD Proxy and 2,073 diagnosed AD). Due to the absence of parental information for some individuals, AD Proxy2 scores were computed for a subset of 355,325 individuals within the aforementioned group. Finally, we adjusted the AD Proxy and AD Proxy2 outcomes for a set of standard covariates: top 20 genetic principal components (PCs), age, and sex. Note that we did not regress out any covariates from the AD outcomes imputed using LS-imputation, PRS-CS, and LDpred2, since those imputation methods predict AD risk based on external GWAS summary statistics that already incorporate covariate adjustment.

## 2.4. TWAS/PWAS data sources

### 2.4.1. Stage 1 data

**GTEx gene expression data** To train transcriptome imputation models, we used genotype data and whole blood gene expression data from GTEx v8, which includes expression data for 19,626 genes [33]. To ensure consistency with the UKB and GWAS data, we subset the GTEx data to individuals with genetically-inferred European ancestry ($n = 558$). We regressed out the effects of 68 covariates provided by GTEx, and used the standardized residuals as the

new gene expression levels. Next, we extracted the local expression quantitative trait loci (*cis*-eQTLs) of each gene from the 100k bp window around its coding region (100k bp upstream from the transcription start site and 100k bp downstream from the transcription end site). In particular, we identified *cis*-eQTLs as follows: Any variants with MAF $\leq$ 0.05 or with any missing values were removed. The remaining local variants for each gene were pruned by removing those with an absolute value of pairwise Pearson correlation $\geq$ 0.8. After pruning, we kept the top 50 *cis*-eQTLs for each gene that are most strongly correlated with its expression.

We applied linear regression with backward selection using AIC as the criterion to build an expression imputation model for each gene. If the resulting model's F-statistic was $\leq$ 10 or the model only had one variant, we excluded that gene from further analysis due to its weak association with the selected *cis*-eQTLs. A total of 3,880 genes had a stage 1 F-statistic $\geq$ 10 and were included in the stage 2 TWAS analysis.

**UKB protein expression data** In addition to TWAS, we also examined associations between the genetic component of protein expression and AD by performing PWAS. To train proteome imputation models, we used imputed genotype data and plasma proteomic data from the UKB, which consists of approximately 36,000 individuals with self-identified white British ancestry (the exact number varies between proteins) and 2,823 proteins coded by autosomal genes [35]. The proteins were mapped to their corresponding genes using UniProt to obtain their *cis*-regions, and then applied quality control steps and local protein quantitative trait loci (*cis*-pQTL) selection procedures analogous to those described above for gene expression data, except that the covariates we adjusted for this time were age, sex, age$^2$, sex*age, sex*age$^2$, and the top 20 genetic PCs. A total of 1,679 proteins had stage 1 F statistic $\geq$ 10 and were included for stage 2 PWAS analysis.

We also considered a different QC process and used LASSO [45] and Elastic-Net [46] as the stage 1 model. Additional analyses using brain hippocampus tissue were also performed with the new stage 1 model and QC process. For full details, see S1 Sect 5.

### 2.4.2. Stage 2 phenotypes

**High-density lipoprotein cholesterol** We considered high-density lipoprotein (HDL) cholesterol as an outcome trait to evaluate the reliability of using an LS-imputed trait to train TWAS stage 2 models. In UK Biobank, missing phenotypes do not pose an issue for HDL. Thus, we imputed HDL trait values only to compare the performance of TWAS methods trained on imputed versus observed traits. By treating the results obtained from training on the observed trait as the ground truth, we could assess the type I error rate and power of TWAS performed with models trained on the imputed trait. For this analysis, we retained individuals with genetically-inferred White British ancestry and no missing HDL values, resulting in a final dataset of 356,351.

**Alzheimer's disease** For our application of TWAS and PWAS to AD, we used data both from the UKB and from release 4 of the ADSP. Stage 2 TWAS models were trained on UKB data with imputed AD status, while association testing was only performed on clinically diagnosed AD cases from ADSP. Depending on the TWAS method used (see Sect 3.3.1), either a portion or the entirety of the ADSP data were reserved for association testing. Note that the ADSP cohorts were designed to ensure an enrichment of AD cases and hence have complete diagnostic information for their participants, so we took those clinical case/control phenotypes as the ground truth and did not perform imputation on the ADSP dataset.

### 2.4.3. Stage 2 datasets

**UK Biobank** We used UKB individuals with imputed AD status as the outcome trait when training stage 2 models for TWAS and PWAS. Namely, we separately considered four imputation strategies: AD status imputed from the EADB GWAS using LS-imputation, AD status

imputed from the IGAP GWAS using LS-imputation, AD status derived from family history using the AD Proxy method, and AD status derived from family history using the AD Proxy2 method. The PRS methods PRS-CS and LDpred2 were not considered in our main analysis due to their poor performance in recovering the genetic architecture of AD (see Sect 3.1 and the S1 Text). A total of 367,182 self-reported White British individuals were included in our analysis, and the same genotype quality control steps and AD status covariate adjustments were applied as previously described in Sect 2.3.3. We set aside 10% of the data as a validation set for early stopping, while the remaining portion was utilized for training the neural network in DeLIVR.

**Alzheimer's Disease Sequencing Project** For sample-level quality control, we followed ADSP recommendations to only keep one sample from each set of genetically-inferred duplicates according to an identical by descent (IBD) analysis ($\hat{\pi} > 0.98$), prioritizing the sample with better sequence call rate or based on reconciliation of phenotypic data. Next, we again followed ADSP recommendations to only keep variants which passed ADSP quality control steps (VFLAGS_One_subgroup = 0) and which have a plausible allele balance of heterozygous calls (ABHet) across samples (0.25 < ABHet < 0.75). This resulted in a total of 314,809,384 high-quality variants, from which we extracted 1,211,080 with MAF $\geq$ 0.05 and Hardy-Weinberg equilibrium exact test $p$-value $\geq$ 0.001. To create a set of harmonized AD case/control phenotypes across the diverse cohorts included in ADSP, we applied the script provided by the ADSP Phenotype Harmonization Consortium (https://github.com/NIAGADS/ADSPIntegratedPhenotypes) with default settings. This yielded 11,308 cases and 16,685 controls for a total sample size of 27,993.

We selected individuals who self-identified as White and Non-Hispanic to match the genetic ancestry of the UKB population. We then excluded individuals with missing age or sex data, leaving 5,857 cases and 4,597 controls for a final sample size of 10,454 individuals. To compute genetic PCs, we first imputed missing genotypes based on allele frequencies and then performed LD clumping to remove variants with a squared correlation higher than 0.2. The top 20 genetic PCs were computed based on the remaining 138,029 variants. Finally, AD status was regressed on several covariates: age, sex, $age^2$, the top 20 genetic PCs, sex*age, and sex*$age^2$.

## 2.5. Simulation

We conducted a simulation study to complement our Type I error control and power analysis. The setup and results are provided in S1 Sect 4.

## 3. Results

In the following sections, we first evaluate the performance of each AD imputation method on GWAS tasks to justify our selection of imputation methods for TWAS/PWAS. We then detail our TWAS analysis workflow for HDL cholesterol and the results we obtained. Note that the target application of our study is AD, for which the number of UKB individuals with available diagnoses is very low. However, we first applied our approach to HDL cholesterol, for which almost all UKB individuals have available measurements, in order to address some key questions regarding the Type I error rate and power of using imputed traits in TWAS/PWAS. In particular, we (a) assessed whether training on imputed traits and testing on observed traits can control the Type I error rate, and (b) compared the power of our imputation-based approach with a TWAS analysis in which stage 2 models are trained and tested on a fully observed trait with limited sample size. Lastly, we present our TWAS and PWAS workflows and analysis results for AD.

## 3.1. Marginal GWAS analysis

For the marginal GWAS analysis, we first imputed AD status for UKB individuals using each of the imputation methods introduced in Sect 2.2, and then performed GWAS with the imputed traits. An additional GWAS analysis was also performed with the diagnosed AD cases for comparison. For the binary AD case/control phenotype obtained using AD Proxy, we fitted a logistic regression model for each variant to obtain its GWAS summary statistics. We then adjusted the resulting marginal effect sizes and standard errors (SEs) by multiplying a factor of 2 as recommended in [10] and [13]. For the continuous AD phenotypes estimated using the other imputation methods, we instead fitted a linear regression model to obtain GWAS summary statistics. To make these summary statistics comparable, we used the formulas in Eqs 7 and 8 to convert the GLM-based summary statistics to OLS-based summary statistics. Here we only provide a brief summary of the results as they are not the main focus of this paper. Additional details can be found in the S1 Text.

### 3.1.1. LS-imputation accurately recovers genetic information in GWAS summary data
We first compared the marginal effect sizes, standard errors (SEs), and $-\log_{10}(p)$-values estimated using imputed AD traits to those reported in the EADB (Figs A-C in S1 Text) and IGAP (Figs D-F in S1 Text) GWAS datasets. Regardless of the GWAS dataset considered, all three quantities estimated with LS-imputed AD status exhibited much higher correlations with the external GWAS data compared to those derived using AD Proxy (Figs G-L in S1 Text). Notably, the correlation between the marginal effect sizes estimated with LS-imputed AD status and the original GWAS estimates reached as high as 0.998, indicating that LS-imputation provides nearly unbiased marginal effect sizes. In contrast, the GWAS effect sizes obtained using AD Proxy outcomes exhibited greater bias.

Next, we visually compared the Manhattan plots of the original GWAS data to the GWAS results obtained using AD status imputed by different methods (Figs M-N in S1 Text). The distribution of significant SNPs identified with LS-imputed AD status (using either IGAP or EADB summary statistics) most closely resembled that obtained from the corresponding GWAS results (IGAP or EADB). The second-best results were obtained with AD Proxy and AD Proxy2. We found that LS-imputation can be more informative than AD Proxy and AD Proxy2, but its performance depends on the sample size and quality of the external GWAS used. Conversely, GWAS analyses performed using AD status imputed by PRS-CS and LDpred2 revealed an inflated number of significant SNPs. This issue was previously highlighted by [15], who noted that any variants included in PRS-CS (or any other PRS) models, along with those in linkage disequilibrium (LD) with them, would be deemed significant given a sufficiently large sample size. We also present Venn diagrams of the significant SNPs identified by different GWAS analyses (Fig O in S1 Text). For this comparison, we considered five different models, excluding those that used PRS-imputed traits due to the inflated numbers of false positives. The conclusions align with those from the Manhattan plots. The distribution of significant SNPs identified using LS-imputed AD status closely resembled that from the corresponding observed GWAS results. Furthermore, LS-imputation appeared to be more informative than AD Proxy or AD Proxy2, depending on the training GWAS data.

### 3.1.2. Comparison of differently imputed AD traits
We explored the similarity of AD traits obtained using different imputation methods (Tables A and B in S1 Text). For every pair of methods, we fitted a linear or a logistic regression model using the AD status from one method as the response and the AD status from the other method as the predictor. Then we calculated the $R^2$ (for linear regression models) or Nagelkerke's $R^2$ (for logistic regression models) for each pair of methods. We noticed that the $R^2$/Nagelkerke's $R^2$ values between AD Proxy and AD Proxy2 were significantly higher than between any other pairs, which

is not surprising as both AD Proxy and AD Proxy2 were constructed mainly with the participants' parental information. The $R^2$ between the two linear imputation methods, PRS-CS and LDpred2, was also higher than between other pairs. For all remaining pairs, the $R^2$/Nagelkerke's $R^2$ values were fairly low, suggesting that the AD outcomes imputed by different methods are nearly uncorrelated with each other. Thus, it might be possible to aggregate information from multiple imputation methods to gain power.

In summary, LS-imputation demonstrated the best performance in GWAS analysis, as both the distribution of significant SNPs and the summary statistics most closely resembled those from the IGAP/EADB GWAS data. Due to their linearity and relatively poorer performance in GWAS, we decided to exclude PRS-CS and LDpred2 from our TWAS and PWAS analyses. Although AD Proxy performed worse than LS-imputation, we retained it in our TWAS/PWAS analyses due to its widespread use for imputing AD cases. However, we excluded AD Proxy2 because of its high correlation with AD Proxy.

## 3.2. Proof-of-concept: TWAS analysis for HDL

### 3.2.1. Analysis workflow for HDL cholesterol
Fig 2 shows our workflow for the TWAS analysis of HDL cholesterol. We created three sets of data for our analysis: $(Z_1, X_1, Y_1)$, $(Z_2, X_2, Y_2)$, and $(Z_2, X_2, \widetilde{Y}_2)$. The first dataset has a sample size of $n_1$ and the latter two datasets each have sample sizes of $n_2$, where $n_1 + n_2 = n = 178,176$. $Y_1$ and $Y_2$ are observed traits, whereas $\widetilde{Y}_2$ is the LS-imputed trait. We will refer to $(Z_1, X_1, Y_1)$ as the observed data, $(Z_2, X_2, \widetilde{Y}_2)$ as the imputed data, and $(Z_1, X_1, Y_1) \cup (Z_2, X_2, Y_2)$ as the complete data. We tested the models on different observed sample sizes, specifically $n_1 = 20,000, 40,000$, and $60,000$.

We developed four distinct approaches to assess the efficacy of the LS-imputation technique based on these three datasets. For the first approach, we divided the observed data $(Z_1, X_1, Y_1)$ into training, validation (for parameter tuning), and testing subsets and used them to train and test the DeLIVR method. The resulting model is denoted as DeLIVR

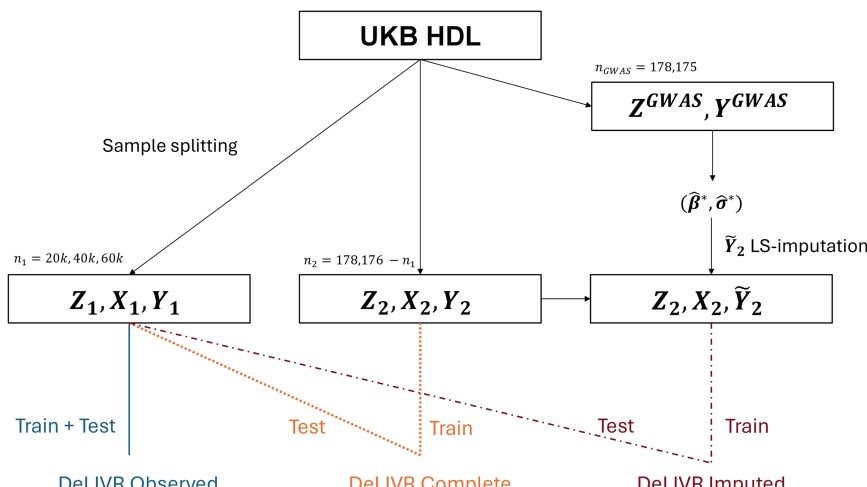

**Fig 2. Flow chart illustrating the TWAS analysis steps for high-density lipoprotein (HDL) cholesterol.** DeLIVR Observed: trained and tested on the observed data with sample splitting; DeLIVR Imputed: trained on the imputed data and tested on the observed data; DeLIVR Complete: trained and tested on the complete data with sample splitting.

Observed. For the second approach, we trained the DeLIVR method using the imputed data $(Z_2, X_2, \widetilde{Y}_2)$ and tested it on the observed data $(Z_1, X_1, Y_1)$. This model is referred to as DeLIVR Imputed. In our third approach, we trained the DeLIVR method using $(Z_2, X_2, Y_2)$ and again tested on the observed data $(Z_1, X_1, Y_1)$. This model is named DeLIVR Complete since it incorporates all available data for both training and testing. Finally, we trained and tested TWAS-LQ on the observed data, which we refer to as TWAS-LQ Observed.

Note that all models were tested on the same set to allow for a direct comparison of the impact of different training data on the association test results. We treated the genes identified by TWAS-LQ Observed, DeLIVR Observed, and DeLIVR Complete as "true positives" and evaluated whether DeLIVR Imputed identified additional genes, which would help assess the Type I error rate of training on imputed traits. We then compared the number of genes discovered by DeLIVR Imputed with those identified by DeLIVR Observed and TWAS-LQ Observed to determine if training with the imputed trait improved power—a key consideration in using imputed traits for TWAS/PWAS.

**3.2.2. LS-imputed HDL enhances power while maintaining type I error rates comparable to real data**   In this section, we evaluate the set of genes identified by DeLIVR when trained on imputed HDL compared to those identified by models trained on observed and complete data, focusing on Type I error control and potential power improvements. Defining true positives in real-data analyses is inherently challenging; thus, our primary aim is to demonstrate that models trained on imputed data maintain Type I error rates comparable to those trained on complete data.

We used the set of genes identified by models trained on observed and complete data as the gold standard. Genes uniquely identified by the model trained on the imputed trait were considered "false positives" introduced by the imputation process. To assess potential power improvements, we examined genes identified using imputed data but not observed data. If these genes were also identified using complete data, this would suggest that training on the imputed trait effectively simulates an increase in sample size, which is the primary motivation for our work.

Fig 3 presents the number of significant genes identified for HDL cholesterol by each TWAS method on a UKB data subset, evaluated across three different sample sizes and two significance cutoffs (Bonferroni and FDR). When the observed data sample size ($n_1$) was 20,000, DeLIVR Imputed discovered two or three more genes than DeLIVR Complete; however, since these genes were also discovered by TWAS-LQ Observed, they are not considered additional false positives introduced by the imputation process. Similar results were observed for $n_1 = 40,000$ and $n_1 = 60,000$, with one exception: at $n_1 = 40,000$, DeLIVR Imputed uniquely identified one or four gene(s) missed by all other methods, which could be potential false positives. In all other cases, DeLIVR Imputed effectively controlled the Type I error rate.

DeLIVR Imputed demonstrated higher power than DeLIVR Observed. Specifically, all genes identified by DeLIVR Observed were also identified by DeLIVR Imputed. Furthermore, when using the Bonferroni cutoff of $1.3 \times 10^{-15}$ (panels a, b, and c), DeLIVR Imputed identified 6 out of 7 and 13 out of 18 genes discovered by DeLIVR Complete at $n_1 = 40,000$ and $n_1 = 60,000$, respectively. The results using an FDR cutoff of 0.1 were similar. These findings indicate that using the LS-imputed trait for training not only improved power compared to training on smaller datasets with observed traits but also maintained a level of power comparable to using the complete dataset. This suggests that the power increase achieved by LS-imputation is akin to a sample size increase, with additional signals identified using imputed traits potentially being detectable with more observed trait data.

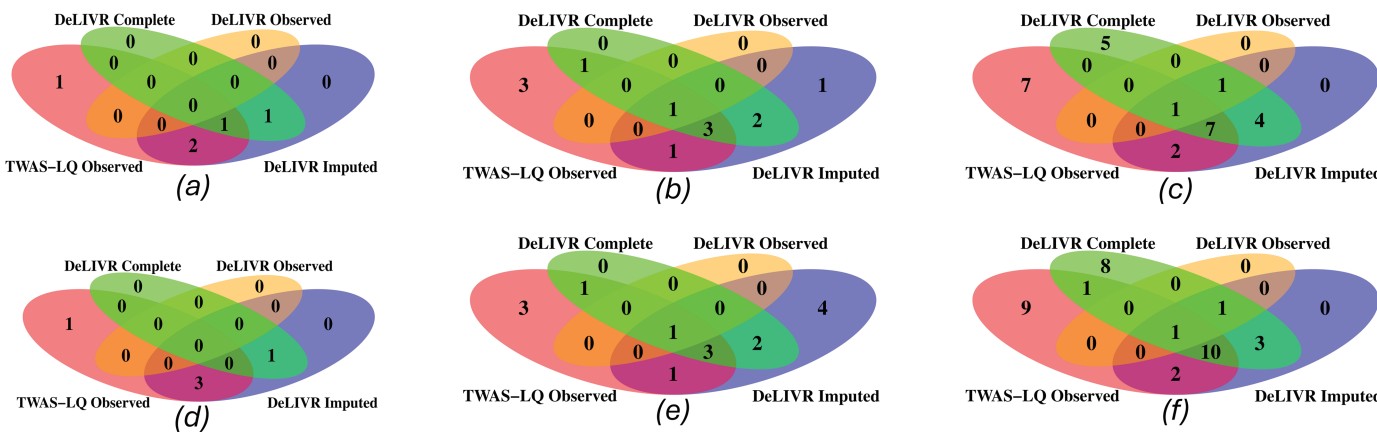

**Fig 3. Venn diagrams showing the numbers of significant genes identified for HDL cholesterol using different observed sample sizes and significance cutoffs.** Top row: Bonferroni cutoff $1.3 \times 10^{-5}$; Bottom row: FDR cutoff 0.1. Sample sizes: (**a**), (**d**): 20k; (**b**), (**e**): 40k; (**c**), (**f**): 60k. DeLIVR Imputed: trained on imputed data and tested on observed data; DeLIVR Observed: trained and tested on observed data with sample splitting; DeLIVR Complete: trained and tested on complete data with sample splitting; TWAS-LQ Observed: trained and tested on observed data with sample splitting.

Finally, we conducted simulations to evaluate Type I error control and power increase as a complement to our real-data analysis. The conclusions were consistent with those from the real-data analysis: DeLIVR Imputed controlled the Type I error rate at the nominal level of 0.05 and demonstrated significantly higher power than DeLIVR Observed. Full details are provided in S1 Sect 4.

### 3.3. TWAS/PWAS analyses for Alzheimer's disease

**3.3.1. Analysis workflow for Alzheimer's disease** In our primary analysis for AD, we trained the stage 2 DeLIVR models on UKB data with imputed AD outcomes and then performed hypothesis testing on clinically diagnosed cases/controls from the ADSP data (Fig 4). Specifically, we first imputed AD status for the UKB individuals using LS-imputation and AD Proxy. For LS-imputation, we created two sets of imputed traits: one using IGAP GWAS summary data and the other using EADB GWAS summary data. We then trained DeLIVR on each of the three imputed UKB datasets and performed association testing on the ADSP dataset. The same workflow and methods were used for both TWAS and PWAS, the only difference being that TWAS relied on transcriptomic data to train expression prediction models in stage 1 while PWAS utilized proteomic data.

In a separate analysis, we considered using the ADSP data for both training and testing, following the methodology from the original DeLIVR paper [36]. For that analysis the ADSP data were divided into training, validation (for parameter tuning), and test sets, to which we applied the DeLIVR method (Fig 4). Our goal was to compare the performance of training on the imputed trait from a large, independent biobank and testing on the real trait against the performance of both training and testing on the real trait. To compare the DeLIVR results with those from a parametric TWAS method, we also applied TWAS-L and TWAS-LQ, which were trained and tested on ADSP data.

All analyses were conducted three times with varying training parameters, and the resulting $p$-values were combined using the Cauchy combination test [47]. Note that association testing was always performed on the observed ADSP data. We intentionally refrained from using imputed AD outcomes for hypothesis testing for two reasons. First, using the imputed

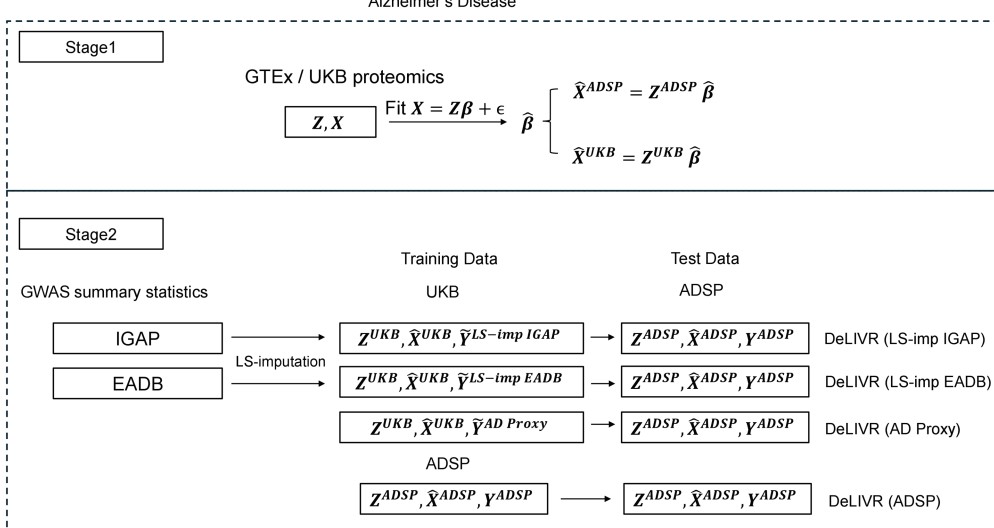

**Fig 4. Flow chart illustrating the TWAS and PWAS analysis steps for AD.** In stage 1, gene expression prediction models for TWAS were trained on GTEx data, and protein expression prediction models for PWAS were trained on UKB data. The model weights were then used to predict expression levels for UKB and ADSP individuals. In stage 2, DeLIVR models were trained using predicted expression levels from stage 1 and one of the following outcome traits: AD status imputed in UKB using LS-imputation and the IGAP GWAS, AD status imputed in UKB using LS-imputation and the EADB GWAS, AD status imputed in UKB using the AD Proxy method, or clinically diagnosed AD status in ADSP. Association testing was performed using the differently trained stage 2 models applied to clinically diagnosed AD status in ADSP individuals.

trait for hypothesis testing in TWAS requires more justification. Second, the LS-imputed trait exhibits correlation among individuals, making it difficult to create an independent test set as required by DeLIVR.

**3.3.2. DeLIVR with imputed AD traits identified Alzheimer's risk genes missed by models trained with observed AD status** Table 1 shows the genes identified by each model. The results focus on the global test (see Sect 2.1.3) since the nonlinear test did not yield any statistically significant findings. The DeLIVR model trained on LS-imputed UKB data using IGAP or EADB summary statistics identified the gene *FNBP1L*, which aligns with recent literature suggesting its association with AD [48]. Similarly, DeLIVR trained on AD Proxy outcomes identified *CR1L*, a gene whose variants have been previously linked to AD risk [5]. Furthermore, the gene *MS4A6A*, well-recognized for its relation to AD, was consistently identified by DeLIVR when trained on LS-imputed data using EADB summary statistics, as well as by both TWAS-L and TWAS-LQ models [49–54]. In contrast, the DeLIVR model trained directly on the observed ADSP data did not identify any of these key genes, possibly due to the much smaller available sample size. As a side note, the well-known AD risk gene *APOE* was not identified, as it did not pass stage 1 QC, regardless of the tissue type or stage 1 model used. It also exhibited a small $R^2$ and a large $p$-value in the precomputed stage 1 model provided by FUSION [19]. Fig 5a shows an UpSet plot summarizing the overlap and uniqueness of genes identified by each method using a $1 \times 10^{-3}$ significance threshold. The full table of $p$-values is provided in Table J in S1 Text. Relaxing the threshold revealed 17 genes uniquely identified by the DeLIVR models. The Q-Q plot (Fig 5b) displays the distribution of $p$-values for the TWAS results across all tested genes. The observed $p$-value distribution closely aligns with the expected null distribution, indicating that the models did not exhibit systematic

**Table 1. $p$-values of the three genes identified by at least one method.** Bonferroni cutoff: $0.05/3880 = 1.3 \times 10^{-5}$. The top row shows the models evaluated. The second row shows the training sets used. All models were tested on the ADSP data. $p$-values smaller than the Bonferroni cutoff are highlighted in bold.

| Model | | DeLIVR | | | | TWAS-L | TWAS-LQ |
|---|---|---|---|---|---|---|---|
| Training data | | ADSP | LS-imp IGAP | LS-imp EADB | AD Proxy | ADSP | ADSP |
| Gene | Chr | | | | | | |
| *FNPB1L* | 1 | 8.5e-2 | **1.2e-5** | **8.3e-6** | 2.6e-5 | 4.7e-2 | 7.1e-3 |
| *CR1L* | 1 | 2.8e-3 | 6.4e-5 | 1.9e-5 | **2.4e-6** | 3.8e-1 | 3.5e-1 |
| *MS4A6A* | 11 | 1.5e-2 | 3.7e-5 | **3.9e-7** | 2.4e-4 | **2.2e-7** | **5.8e-7** |

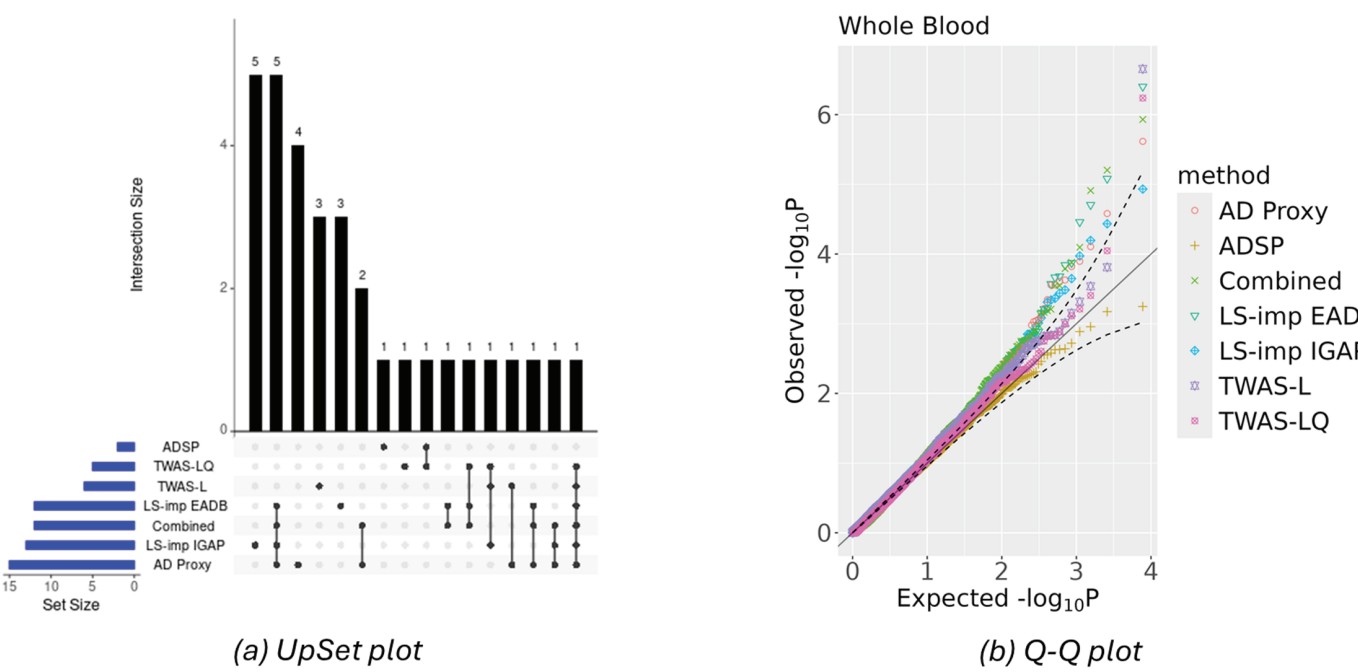

*(a) UpSet plot*                     *(b) Q-Q plot*

**Fig 5. Side-by-side UpSet plot and Q-Q plot of all models for TWAS.** "ADSP" refers to the DeLIVR model trained on the ADSP data (observed AD status). "TWAS-L" and "TWAS-LQ" refer to the standard TWAS model and the parametric TWAS-LQ model trained on the ADSP data, respectively. "LS-imp EADB" and "LS-imp IGAP" refer to the DeLIVR model trained on the LS-imputed AD status using EADB and IGAP as the GWAS data, respectively. "AD Proxy" refers to the DeLIVR model trained on the proxy AD status. "Combined" refers to the results of the Cauchy combination test, which combines the $p$-values from "LS-imp EADB," "LS-imp IGAP," and "AD Proxy."

inflation of the test statistics. These results underscore the efficacy of using imputed traits to uncover nonlinear genetic associations with AD that might be missed by direct analysis of limited observed data alone.

Additional analyses using different stage 1 QC criteria and models were conducted for both whole blood and brain hippocampus tissues, as detailed in S1 Sect 5. The conclusions remained largely unchanged with one notable exception: the Cauchy combination test identified more genes than using any single set of imputed trait values. This result is likely due to the distinctive gene sets identified by each set of imputed trait values.

**3.3.3. PWAS performed on imputed Alzheimer's status identified known and putative risk proteins** Table 2 shows the significant proteins identified by at least one PWAS method. Across all models, the well-established AD risk proteins apolipoprotein E (APOE) and apolipoprotein C-I (APOC1) were consistently identified. CR1 is another protein known to be associated with AD, which was identified by DeLIVR trained on AD status imputed

**Table 2. *p*-values of the significant proteins identified for Alzheimer's disease by at least one method.** Bonferroni cutoff: **0.05/1679** = $2.98 \times 10^{-5}$. The top row shows the models evaluated. The second row shows the training sets used. All models were tested on the ADSP data. *p*-values smaller than the Bonferroni cutoff are highlighted in bold.

| Model | | DeLIVR | | | | TWAS-L | TWAS-LQ |
|---|---|---|---|---|---|---|---|
| Training data | | ADSP | LS-imp IGAP | LS-imp EADB | AD Proxy | ADSP | ADSP |
| **Protein name** | **Gene name** | | | | | | |
| Apolipoprotein E | *APOE* | **2.8e-16** | **0** | **0** | **0** | **1.7e-44** | **2.9e-44** |
| Apolipoprotein C-I | *APOC1* | **3.2e-6** | **1.5e-10** | **4.3e-8** | **2.1e-8** | **5.0e-7** | **3.3e-6** |
| Complement receptor type 1 | *CR1* | 1.9e-3 | 3.4e-2 | **2.0e-5** | 3.4e-3 | **2.4e-6** | **1.0e-5** |
| Basal cell adhesion molecule | *BCAM* | 1.8e-3 | **1.9e-6** | 3.2e-4 | **3.2e-6** | **1.7e-9** | **1.0e-24** |
| Lipocalin-15 | *LCN15* | **2.1e-5** | 1.7e-3 | 5.2e-4 | 1.4e-3 | 1.8e-4 | 8.8e-4 |
| Carbonic anhydrase 13 | *CA13* | 3.5e-2 | 7.8e-3 | 1.1e-2 | 1.1e-2 | 6.9e-4 | **3.3e-7** |
| Retinoic acid receptor responder protein 2 | *RARRES2* | 1.8e-5 | **1.2e-5** | 2.2e-4 | **1.9e-6** | 9.8e-4 | 4.4e-3 |

by LS-imputation with EADB GWAS data, as well as by the two parametric models. BCAM was also identified by both parametric models, as well as by DeLIVR when trained on AD outcomes imputed by AD Proxy and by LS-imputation with IGAP summary statistics. RARRES2 was uniquely identified by DeLIVR trained on imputed AD outcomes, namely when we used the imputation methods of AD Proxy and LS-imputation with IGAP GWAS data. Recent research has suggested that this protein may act as a risk factor for AD [55,56], supporting the potential validity of our findings. Interestingly, LCN15 was uniquely identified by DeLIVR trained on the observed ADSP data, despite its smaller sample size. There was also one protein, CA13, that was uniquely identified by TWAS-LQ. Fig 6a presents the UpSet plot using a significance cutoff of $1 \times 10^{-3}$. Unlike the TWAS results, the parametric models using the observed AD status identified the highest numbers of proteins. However, DeLIVR uniquely identified a number of proteins, complementing the models trained on the observed AD status. Fig 6b displays the Q-Q plot for all proteins. The distributions of *p*-values exhibit a similar pattern across all models.

## 4. Discussion

The results of this study underscore the potential of proxy phenotypes and LS-imputation to address the challenges posed by the small number of AD cases in large biobank data, which otherwise limits the discovery of genetic signals related to AD. By leveraging the proxy AD and LS-imputation methods, we observed consistent and reliable GWAS results across different datasets. Specifically, the comparison of GWAS summary statistics derived from imputed AD traits using either proxy AD or LS-imputation with those from other popular methods highlighted the robustness of these two approaches in capturing the genetic underpinnings of AD. However, proxy AD and GWAX cannot be applied without parental/sibling information, while LS-imputation can gain information from external large-scale GWAS summary data. Overall, we found that trait imputation, e.g., via proxy AD and LS-imputation, proved effective in overcoming the limitations posed by the sparsity of AD cases in biobank data, thereby broadening the scope of genetic research in AD.

Furthermore, the application of DeLIVR, a nonlinear TWAS method, to LS-imputed AD status provided novel insights into the genetic landscape potentially driving AD pathology. The identification of significant genes and proteins linked to AD, such as the gene *FNPB1L* and protein CR1, underscores the method's utility in identifying biologically relevant targets. Although DeLIVR trained on the imputed traits did not always identify the largest number

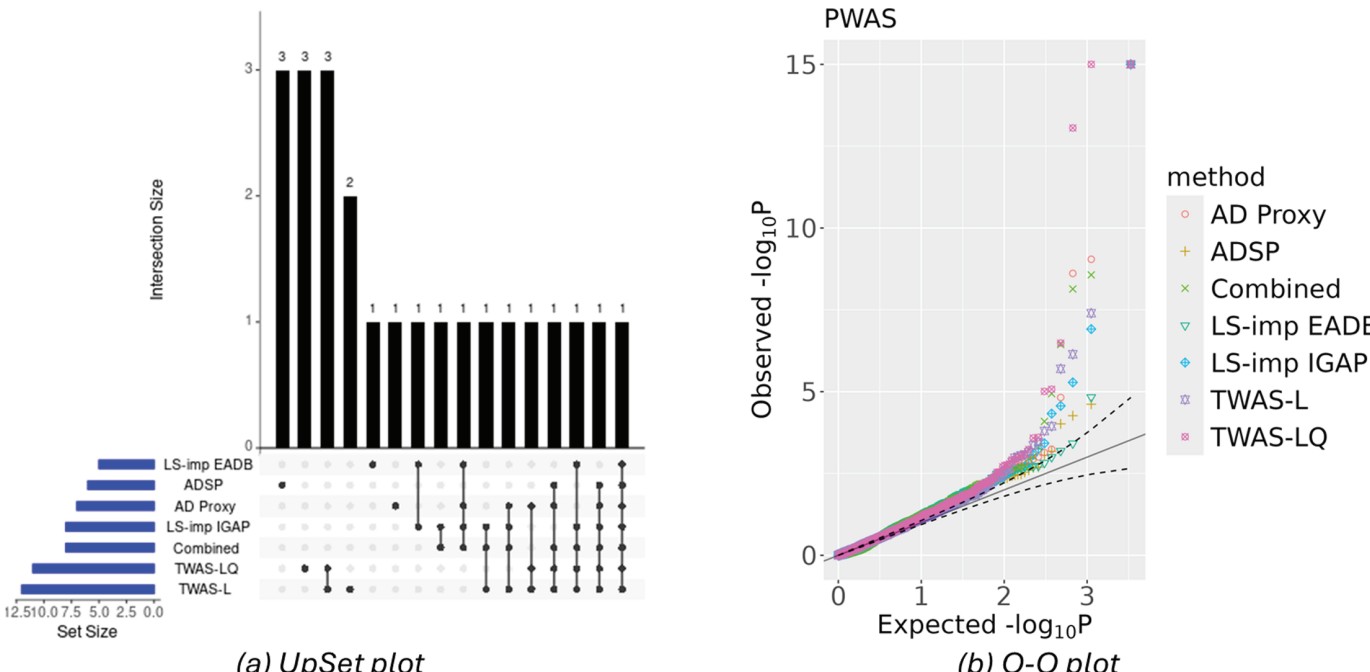

**Fig 6. Side-by-side UpSet plot and Q-Q plot of all models for PWAS.** "ADSP" refers to the DeLIVR model trained with the ADSP data (the observed AD status). "TWAS-L" and "TWAS-LQ" refer to the standard TWAS model and the parametric TWAS-LQ model trained with the ADSP data, respectively. LS-imp EADB and LS-imp IGAP refer to the DeLIVR model trained with the LS-imputed AD status using EADB and IGAP as the GWAS data, respectively. "AD Proxy" refers to the DeLIVR model trained with the proxy AD status. "Combined" refers to the results of the Cauchy combination test combining the $p$-values of "LS-imp EADB," "LS-imp IGAP," and "AD Proxy."

of genes or proteins, it uniquely identified some genes and proteins that would be missed using conventional GWAS or TWAS, highlighting the method's added value. This is particularly relevant given that nonlinear associations may be preserved by LS-imputation, but not by traditional linear PRS models.

Our findings also suggest that different imputation methods can complement each other, particularly when the methods exhibit significant distinctiveness, revealing a broader spectrum of genetic influences on AD that could be missed when relying on a single method. This point is further supported by additional analyses performed on brain hippocampus tissue, where aggregating results from different imputation methods yielded the largest set of genes discovered. These complementary effects have the potential to enhance the overall power of genetic studies, underscoring the value of employing a multifaceted approach to trait imputation.

We acknowledge several limitations that warrant further investigation. First, the computational complexity of LS-imputation limits the number of genetic variants that can be used. Second, at present, we are unable to perform hypothesis testing on LS-imputed traits for non-parametric TWAS models for two main reasons: 1) the imputed trait values are correlated across batches, which complicates the creation of an independent test set, and 2) testing using the imputed traits could lead to biased results, which is also expected to be a general issue for any trait imputation method. There is always a bias-variance trade-off when using imputed traits; the question of how to best balance this trade-off remains largely open. Third, our analysis was limited to individuals of European ancestry. A multi-ancestry design would be necessary to ensure that the findings are generalizable across diverse populations [57]. Fourth, our

analysis focused solely on whole blood tissue, but aggregating data from multiple tissues could provide additional insights and improve the robustness of the findings [58,59].

In conclusion, LS-imputation represents a significant advance in the field of genetic research into AD, providing a robust tool for enhancing the quality of phenotypic data and, consequently, the insights gleaned from genetic association studies. Future research should focus on refining these imputation techniques and exploring their applications across other complex diseases to fully realize their potential in precision medicine and genetics research.

## Supporting information

**S1 Text. Supplementary file with details of additional quality control procedure, real data analysis results, and simulation results.**
(PDF)

## Acknowledgments

Access to the GTEx data was approved for dbGaP Project #26511, and the data were obtained from dbGaP accession number phs000424.v8.p2 on 10/13/2021. Access to the UK Biobank (UKB) data was approved through UKB Application #35107. Data from the Alzheimer's Disease Sequencing Project (ADSP) were prepared, archived, and distributed by the National Institute on Aging Alzheimer's Disease Data Storage Site (NIAGADS) at the University of Pennsylvania. The authors also acknowledge the Minnesota Supercomputing Institute (MSI) at the University of Minnesota for providing high-performance computing resources that contributed to the research results reported within this paper. We thank the International Genomics of Alzheimer's Project (IGAP) for providing summary results data for these analyses. The investigators within IGAP contributed to the design and implementation of IGAP and/or provided data but did not participate in analysis or writing of this report. IGAP was made possible by the generous participation of the control subjects, the patients, and their families.

## Author contributions

**Conceptualization:** Wei Pan.

**Data curation:** Ruoyu He, Jingchen Ren, Mykhaylo M. Malakhov.

**Formal analysis:** Ruoyu He, Jingchen Ren, Wei Pan.

**Funding acquisition:** Wei Pan.

**Investigation:** Ruoyu He, Jingchen Ren, Mykhaylo M. Malakhov, Wei Pan.

**Methodology:** Ruoyu He, Jingchen Ren, Mykhaylo M. Malakhov, Wei Pan.

**Project administration:** Wei Pan.

**Resources:** Wei Pan.

**Software:** Ruoyu He, Jingchen Ren.

**Supervision:** Wei Pan.

**Validation:** Ruoyu He, Jingchen Ren, Mykhaylo M. Malakhov.

**Visualization:** Ruoyu He, Jingchen Ren.

**Writing – original draft:** Ruoyu He, Jingchen Ren.

**Writing – review & editing:** Ruoyu He, Jingchen Ren, Mykhaylo M. Malakhov, Wei Pan.

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
