## [Decision Letter · Decision Letter 0]

7 Oct 2024

Dear Dr Pan,

Thank you very much for submitting your Research Article entitled 'Enhancing nonlinear transcriptome- and proteome-wide association studies via trait imputation with applications to Alzheimer's disease' to PLOS Genetics.

The manuscript was fully evaluated at the editorial level and by independent peer reviewers. The reviewers appreciated the attention to an important problem, but raised some substantial concerns about the current manuscript. Based on the reviews, we will not be able to accept this version of the manuscript, but we would be willing to review a much-revised version. We cannot, of course, promise publication at that time.

If you decide to revise the manuscript for further consideration at PLOS Genetics, please aim to resubmit within the next 60 days, unless it will take extra time to address the concerns of the reviewers, in which case we would appreciate an expected resubmission date by email to plosgenetics@plos.org.

To resubmit, log into your Editorial Manager account and select the option 'Revise Submission' in the 'Submissions Needing Revision' folder.

We are sorry that we cannot be more positive about your manuscript at this stage. Please do not hesitate to contact us if you have any concerns or questions.

Yours sincerely,

Hae Kyung Im

Guest Editor

PLOS Genetics

Xiaofeng Zhu

Section Editor

PLOS Genetics

Please address the reviewer comments.

Reviewer's Responses to Questions

**Comments to the Authors:**

Reviewer #1: He and Ren et. al. present an interesting approach to TWAS and PWAS, asking if outcome trait imputation improves TWAS/PWAS performance, especially for Alzheimer’s Disease (AD), where the number of cases available in large biobanks is limited. They model the omics trait effects on outcome explicitly, focusing on their DeLIVR deep learning model, and compare predicted outcomes to observed outcomes in an independent cohort rather than testing predicted omics traits for association with outcomes like in traditional TWAS. While the authors do successfully demonstrate that by imputing outcomes (HDL or AD) and using DeLIVER, they can detect potentially nonlinear associations between omics traits (exposures) and outcomes, I’m not convinced this method uncovers new gene/protein associations that would not be found in traditional linear TWAS. Clarifications and additions that could improve the paper include:

1. In this paper, you use your HDL analyses to justify using case/control imputation in your AD analyses. As genetic architectures vary among traits, showing similar results with additional traits likely to be present in biobanks, e.g. height, blood pressure, etc., would be more convincing than just HDL. More concerning is your conclusions are based on a handful of significant observations in the Venn diagrams presented in Fig 3. Just because an association was found with TWAS-LQ does not mean it is a “true positive”. I know defining “true positives” is a challenge, but I suggest also calculating the proportion of genes/proteins you discover that are also listed in the GWAS Catalog for a particular trait (HDL, AD, etc.) would be helpful. Additionally, you could perform your intersection analyses at a more relaxed threshold, e.g. FDR<0.1 rather than Bonferroni, since your Venns have such small numbers.

2. The results presented in Tables 1 and 2 seem highly correlated across models, even though only the bolded p-values reach Bonferroni adjustment for the numbers of genes or proteins tested. Indeed, many p-values miss this significance threshold, but are still low. The only unique DeLIVR result I see is CR1L in Table 1. Thus, I am not convinced the associations are unique to deep learning methods. Similar to above, if you relax thresholds, how unique are your DeLIVR results?

3. Along these lines, you tested just 3,880 genes (GTEx whole blood) and 1,679 proteins (UKB plasma) in your models. I’m guessing APOE was not tested in your gene expression (eQTL) models? How many known AD genes were tested in TWAS and PWAS? GTEx also has several brain tissues available that may be quite useful for AD, which I suggest including. Rather than building your own gene expression models, you could use models publicly available for PrediXcan or FUSION to test more genes in your TWAS, ~13k per tissue, which may allow better assessment of potential differences between methods.

4. “LD contamination” is often a problem with TWAS results because genes linked to each other may share eQTLs (Barbeira et al 2018, Zhou et al 2020). Including chromosome and positions in your tables listing genes (and including this info for the Fig 3 results) would be helpful. In addition, colocalization or joint analyses would help zero in on the causal gene/protein just like fine mapping for SNPs.

5. I thought Fig 1 was very clear, thank you. Could you add sample size and sample splitting information to Fig 2 to make it clearer? I suggest UpSet plots rather than Venn diagrams for Fig 3, especially if you end up with larger numbers.

Reviewer #2: Please see the review report.

Reviewer #3: This study evaluates various (Alzheimer's disease) imputation methods in capturing genotype-phenotype relationships and compares these methods in non-linear TWAS/PWAS inference. The framework differs from traditional TWAS in that it requires 3 datasets: 1) a reference dataset to build TWAS/PWAS prediction models 2) a biobank data with genotype data but without phenotypes (Using a trait imputation method to impute the outcome into this biobank and using imputed expression from stage 1, the framework then trains a stage 2 model using the predicted expression as input and imputed trait as output.) 3) Finally, an independent test set with observed genotype and observed outcome trait data, which is used for hypothesis testing to test the association between predicted outcome trait and its observed values. The framework uses a neural network approach, DeLIVR, to nonparametrically estimate E[Y | genotype] in stage 2. The authors analyzed several trait imputation methods: LS-imputation, Proxy-AD (of 2 types), PRS-CS (where the risk score is used as the imputed trait), and LDpred2 (where again the risk score is used as the imputed trait). GTEx and UKB were the sources of the reference transcriptome/proteome data for training models of these molecular traits.

Overall, this study adds to the TWAS literature and offers some important methodological insights. The systematic benchmarking of the various methods (at each stage) will be useful to the community. Overall, the paper is well-written. However, I do have some concerns, which I think should be addressed.

1. Please connect the comparison of marginal effect sizes, standard errors, and p-values reported in Results to the actual tables/figures in Supplementary Information. It was challenging to follow the results.

2. How was the distribution of significant SNPs identified via LS-imputation (and the other trait imputation methods) compared to the results from the corresponding GWAS results (IGAP or EADB). The authors indicate that they "compared the Manhattan plots" of the original GWAS to the results from the trait imputation methods. There's a lot of hand-waving in this section of results and quite a bit of subjectivity.

3. Similarly for the comparison of the various imputed traits (using R2 or Nagelkerke’s R2) in Results, it would be important to link this section of Results to the relevant Supplementary Information.

4. I'm concerned that the TWAS did not identify the well-known Alzheimer's gene APOE, which has been consistently identified by previous TWAS studies (and implicated by previous GWAS). Now, of course, APOE was identified in the PWAS analysis (on imputed AD status). How do the authors interpret these findings?

5. Given that with the exception of APOE and APOC1, no other proteins were identified by DeLIVR across all imputation approaches, how specifically would the authors "combine" the various results? It's unclear how this combination should be done; it would be useful if the authors can make recommendations along these lines given the results presented here.

6. The study was done using GTEx whole blood. The tissue specificity of the approach/results is unclear. (There's a strong argument to be made that whole blood is not the ideal tissue for the analysis, given the availability of brain in GTEx (and other reference resources) and given its well-recognized cell type heterogeneity.)

7. The study started with >2800 proteins, but ended up with only <1700 proteins for the stage 2 PWAS analysis. It's unclear how many proteins were dropped at each step of the preprocessing and QC. This would be very useful information.

8. It would be important to present the Q-Q plot of the association p-values. It's important for readers to see this distribution given the control for type I error claimed here.

9. The authors should make all results available, including the TWAS/PWAS models used and the association results (not just the code), for ease of reproducibility.

**Have all data underlying the figures and results presented in the manuscript been provided?**

Reviewer #1: None

Reviewer #2: None

Reviewer #3: Yes

PLOS authors have the option to publish the peer review history of their article (what does this mean?). If published, this will include your full peer review and any attached files.

Reviewer #1: No

Reviewer #2: No

Reviewer #3: **Yes: **Eric R. Gamazon

---

## [Decision Letter · Decision Letter 1]

26 Jan 2025

PGENETICS-D-24-00954R1

Enhancing nonlinear transcriptome- and proteome-wide association studies via trait imputation with applications to Alzheimer's disease

PLOS Genetics

Dear Dr. Pan,

Thank you for submitting your manuscript to PLOS Genetics. After careful consideration, we feel that it has merit but does not fully meet PLOS Genetics's publication criteria as it currently stands. Therefore, we invite you to submit a revised version of the manuscript that addresses the points raised during the review process.

Please submit your revised manuscript within 60 days Mar 27 2025 11:59PM. If you will need more time than this to complete your revisions, please reply to this message or contact the journal office at plosgenetics@plos.org. Please include the following items when submitting your revised manuscript:

We look forward to receiving your revised manuscript.

Kind regards,

Hae Kyung Im

Guest Editor

PLOS Genetics

Xiaofeng Zhu

Section Editor

PLOS Genetics

Aimée Dudley

Editor-in-Chief

PLOS Genetics

Anne Goriely

Editor-in-Chief

PLOS Genetics

**Additional Editor Comments :**

Apologies for the delay in the decision. However, we would like to see the issue on type I errror raised by reviewer #2 to be addressed before making a decision.

**Journal Requirements:**

1) Thank you for stating that "Individual-level data from the UK Biobank (UKB {https://www.ukbiobank.ac.uk/}), the Genotype-Tissue Expression (GTEx {https://gtexportal.org/home/}) Project, and the Alzheimer's Disease Sequencing Project (ADSP {https://adsp.niagads.org/}) are available by application through their respective data access processes." Please note that these links seem to direct to very general homepages rather than pages with the specific data. Please update your statement to provide further details or direct page links.

2) The file inventory includes files for Figures 3a, 3b,3c,3d,3e,3f, 5a,5b,6a and 6b. We would recommend either combining these into single Figure 3.tiff, Figure 5.tiff and Figure 6.tiff files with separate internal panels, or renumbering them as individual figures, as we are not able to publish multiple components of a single figure as separate files.

3) Please ensure that the affiliations of the authors listed on the manuscript title page do exactly match with the affiliations provided in the online submission form.

NOTE: Affiliations should include a department (if applicable), an institution, a city, and a country.

**Reviewers' comments:**

Reviewer's Responses to Questions

Reviewer #1: The authors have addressed most of my concerns. I have a few minor suggestions given the additional analyses conducted:

1. The Discussion limitations have not been updated to include your brain hippocampus analysis. This analysis should be mentioned in the main text.

2. Similarly, your response to reviewers about APOE transcript models should be included in the main text or supplement as readers will want to know this information.

3. Thank you for including your code in github. Please be sure to add a README to describe what scripts and results files are included.

Reviewer #2: For my previous comments 1&2, 4 (Type I error):

The authors have added additional simulations to address my previous comments. However, the type I error analysis remains unconvincing. It does not seem reasonable to use other existing methods as a gold standard to determine whether a new discovery is a false positive. For instance, the authors state that in one case (n_1 = 4000), DeLIVR Imputed uniquely identified several genes missed by all other methods, suggesting these could be potential false positives. This raises concerns about the claim that DeLIVR Imputed has higher power. As a result, this approach does not appear to provide a valid method for demonstrating control of type I error.

For my previous comments 3&6:

There seems to be a contradiction in the authors’ responses. In response to my comment 3, the authors stated: “First, our main point is not about the combination of methods, but that trait imputation can improve over using small samples of observed traits only.” However, in response to comment 6, they stated: “It is not the main point of the current paper to investigate the performance of DeLIVR, but the combined use of DeLIVR and trait imputation.” These statements appear inconsistent, leaving me unclear about the main purpose of the paper. Please clarify the primary focus and goals in the manuscript to address this confusion.

Then, in response to the author’s comments to my comment 6: The presentation of the results is somewhat confusing and gives the impression that the authors are claiming DeLIVR combined with other methods is superior to TWAS-L and TWAS-LQ. However, based on the data analysis results, particularly Table 2, I still do not see a clear advantage of DeLIVR combined with other imputation methods compared to existing TWAS approaches (e.g., TWAS-L and TWAS-LQ). If this is not the intended message, please clarify this in the manuscript.

**Have all data underlying the figures and results presented in the manuscript been provided?**

Reviewer #1: None

Reviewer #2: None

PLOS authors have the option to publish the peer review history of their article (what does this mean?). If published, this will include your full peer review and any attached files.

Reviewer #1: No

Reviewer #2: No

**Figure resubmission:**
---

## [Decision Letter · Decision Letter 2]

18 Mar 2025

Dear Dr Pan,

We are pleased to inform you that your manuscript entitled "Enhancing nonlinear transcriptome- and proteome-wide association studies via trait imputation with applications to Alzheimer's disease" has been editorially accepted for publication in PLOS Genetics. Congratulations!

Yours sincerely,

Hae Kyung Im

Guest Editor

PLOS Genetics

Xiaofeng Zhu

Section Editor

PLOS Genetics

Aimée Dudley

Editor-in-Chief

PLOS Genetics

Anne Goriely

Editor-in-Chief

PLOS Genetics

Comments from the reviewers (if applicable):

Reviewer's Responses to Questions

**Comments to the Authors:**

Reviewer #1: The authors have satisfactorily addressed my concerns.

Reviewer #2: The authors addressed my concerns.

**Have all data underlying the figures and results presented in the manuscript been provided?**

Reviewer #1: None

Reviewer #2: Yes

PLOS authors have the option to publish the peer review history of their article (what does this mean?). If published, this will include your full peer review and any attached files.

Reviewer #1: No

Reviewer #2: No

**Data Deposition**

http://datadryad.org/submit?journalID=pgenetics&manu=PGENETICS-D-24-00954R2

**Press Queries**

---

## [Editor Report · Acceptance letter]

PGENETICS-D-24-00954R2

Enhancing nonlinear transcriptome- and proteome-wide association studies via trait imputation with applications to Alzheimer's disease

Dear Dr Pan,

We are pleased to inform you that your manuscript entitled "Enhancing nonlinear transcriptome- and proteome-wide association studies via trait imputation with applications to Alzheimer's disease" has been formally accepted for publication in PLOS Genetics! Your manuscript is now with our production department and you will be notified of the publication date in due course.

With kind regards,

Lilla Horvath

PLOS Genetics

On behalf of:
